# VoiceBlock: Privacy through Real-Time Adversarial Attacks with Audio-to-Audio Models

**Patrick O'Reilly, Andreas Bugler, Keshav Bhandari, Max Morrison, Bryan Pardo**
Department of Computer Science, Northwestern University
{patrick.oreilly2024, andreas, keshavbhandari2023}@u.northwestern.edu
pardo@northwestern.edu

## Abstract

As governments and corporations adopt deep learning systems to collect and analyze user-generated audio data, concerns about security and privacy naturally emerge in areas such as automatic speaker recognition. While audio adversarial examples offer one route to mislead or evade these invasive systems, they are typically crafted through time-intensive offline optimization, limiting their usefulness in streaming contexts. Inspired by architectures for audio-to-audio tasks such as denoising and speech enhancement, we propose a neural network model capable of adversarially modifying a user's audio stream in real-time. Our model learns to apply a time-varying finite impulse response (FIR) filter to outgoing audio, allowing for effective and inconspicuous perturbations on a small fixed delay suitable for streaming tasks. We demonstrate our model is highly effective at de-identifying user speech from speaker recognition and able to transfer to an unseen recognition system. We conduct a perceptual study and find that our method produces perturbations significantly less perceptible than baseline anonymization methods, when controlling for effectiveness. Finally, we provide an implementation of our model capable of running in real-time on a single CPU thread. Audio examples and code can be found at https://interactiveaudiolab.github.io/project/voiceblock.html.

## 1   Introduction

Mass surveillance of voice communications is an ongoing and pervasive issue. While section 702 of the United States Foreign Intelligence Surveillance Act allows the government to perform targeted monitoring of foreign communications, bulk collection practices have resulted in the warrantless surveillance of large numbers of "incidental" foreign and domestic individuals [17]. Despite the fact that millions of these communications are obtained without warrants, they have been used in ordinary criminal investigations [60], undermining a core purpose of the Fourth Amendment of the US constitution [59]: to protect the people against searches without probable cause. Many corporations also possess the capability for large-scale collection of voice data [23], which may be leveraged to profile users for advertising or accessed by government entities through upstream and downstream surveillance [16]. As individuals are faced with these growing surveillance apparati, it is worth remembering that routine surveillance of private voice communications is also corrosive to free speech and association and tends to disproportionately affect marginalized groups [28, 18].

In the absence of identifying metadata, automatic speaker recognition systems can facilitate mass surveillance by allowing an operator to search a database of recorded voice data for utterances from a chosen speaker [7] or to diarise (assign utterances to individuals) transcripts of recordings [44]. Prior to the advent of automatic speaker recognition these tasks required human analysts, forming a natural check on surveillance overreach. We seek to restore this check by degrading the efficacy of speaker

36th Conference on Neural Information Processing Systems (NeurIPS 2022).

recognition models while maintaining the original perceptual quality of the voice communication, a step that could grant users a measure of privacy from mass surveillance.

Modern speaker recognition systems rely on deep neural networks [6]. Deep networks have been shown to be vulnerable to adversarial examples—natural instances (e.g. a recording of "Bob" speaking) modified to cause a model to make an incorrect prediction (the recording's speaker is labeled as "Maria") [53]. This presents opportunities for privacy-minded individuals to mislead or evade surveillance systems with adversarially-crafted inputs. Researchers have proposed adversarial attacks against a variety of audio systems, including speaker recognition [62, 51].

To fool a given neural network-based system, many audio attacks modify a recording by adding a perturbation signal directly at the waveform representation [8]. This perturbation is typically crafted using gradient information obtained from the system's neural network—or from a similarly-constructed surrogate—and requires iterative optimization. Once the optimization is complete, the modified recording is played to the system in an effort to induce arbitrary incorrect predictions (an *untargeted* attack) or a specific incorrect prediction chosen by the attacker (a *targeted* attack).

In theory, such attacks might allow individuals to evade systems that surveil and analyze voice communications, thereby protecting privacy. However, many existing attack algorithms require a costly optimization for each audio recording to which they are applied. Continuous, real-time voice communication precludes the adoption of such approaches through online optimization (which is typically too slow) or through the use of a set of precomputed adversarial examples (which would necessarily limit the user's interaction). This suggests a need for algorithms capable of modifying speech on-the-fly.

Recent years have seen the development of models for audio-to-audio tasks such as denoising, voice conversion, and musical timbre transfer [13, 48, 14]. Such models have sufficient expressive power to modify audio in coherent and task-specific ways, and are often designed to run in real-time. Inspired by these models, we propose **VoiceBlock**, a deep network that learns to apply a time-varying finite impulse response (FIR) filter to outgoing audio, producing highly effective adversarial perturbations on a small fixed delay suitable for streaming. The main contributions of this work are:

- A highly effective method for de-identifying speech in real-time that is more perceptually inconspicuous than existing methods of equal effectiveness
- Objective and subjective experimental results that validate these claims
- A system that embodies the claimed advances, and that runs in real-time on a single CPU thread

Through this work, we hope to encourage further exploration of audio-to-audio models for protecting user privacy. Audio examples and code, including our streaming implementation, can be found at `https://interactiveaudiolab.github.io/project/voiceblock.html`.

## 2 Related work

Previous works in **speech de-identification** have proposed methods for obfuscating speech to evade surveilling speaker-recognition systems. One such method is voice conversion, which modifies the speech of a source speaker to sound like that of another target speaker, both to humans and machines. Jin et al. [24] and Alegre et al. [4] analyze the vulnerability of speaker recognition systems against de-identification attacks performed with voice conversion models. However, the conversion models considered are incapable of real-time operation. While the recent availability of low-latency voice conversion models [48, 49, 31] may make such approaches more practical, our aim is to modify speech in a way that is inconspicuous and minimally invasive to the user experience, and that has the potential to generalize to tasks beyond speaker recognition. This also rules out recent anonymization approaches catalogued through the VoicePrivacy Challenge, as all evaluated submissions to date noticeably alter speaker characteristics [55]. Similarly, we do not consider obvious transformations like pitch-shifting that significantly alter or degrade recorded audio.

**Real-time audio attacks** have recently been explored. Chiquier et al. [10] propose an attack on automatic speech recognition systems that is capable of inducing significant transcription errors in an *over-the-air* environment (where attack audio is played to the system in a physical space), but which introduces conspicuous noise due to the use of additive perturbations at the audio waveform. Their

approach also necessitates both a large model (requiring approximately 16 GPU-days to train) and lengthy receptive field, resulting in an initial "idle" period of 2.5s during which the attack does not affect speech. By contrast, we focus on an *over-the-line* setting, where attacks are passed to the victim model over a digital channel, and propose a lightweight network capable of producing inconspicuous perturbations through the use of filtering, with an initial delay of miliseconds rather than seconds.

Universal adversarial attacks optimize a short perturbation that can be played alongside arbitrary speech, optionally in real-time, to evade a victim model [32, 62, 27]. However, these attacks tend to introduce conspicuous noise due to the use of additive perturbations at the audio waveform. The same holds for the attack of Xie et al. [61], which uses a Wave-U-Net [52] model to efficiently generate universal perturbations. Rather than generate additive perturbations, we perform multiplicative attacks in the frequency domain through the use of filtering, and in doing so avoid the bias towards noise-like artifacts typical of additive attacks.

Other works have proposed crafting **adversarial attacks in the frequency domain**. The "Kenansville" attack of Abdullah et al. [1] performs spectral gating, removing low-energy frequency components in an effort to hinder classification. The authors demonstrate the effectiveness of their approach against seen and unseen speaker- and speech-recognition systems, and the proposed method is fast and gradient-free. However, the spectral gating must be constrained to a small number of key audio frames to avoid conspicuous artifacts, which requires offline optimization using word- or phoneme-level alignment information.

Ahmed et al. [3] optimize bandpass filters to perform targeted impersonation attacks on speaker-recognition models; these filters can then be physically realized as resonant tubes through which an attacker can speak to the victim system, allowing real-time operation. However, once optimized and realized, the bandpass filters are fixed in time and cannot adapt to the attacker's speech. The attack is significantly less effective than traditional approaches, even in controlled acoustic environments, and requires an active intervention on the part of the user to both realize and apply.

Finally, O'Reilly et al. [41] adversarially optimize the parameters of a time-varying finite impulse response (FIR) filter in order to avoid noisy artifacts. However, the attack operates offline and must be optimized separately for every instance to which it is applied.

In summary, we are aware of no current approach that is simultaneously capable of performing de-identification while remaining inconspicuous to human listeners and operating in real-time on-device.

## 3 VoiceBlock

We propose VoiceBlock, a system that applies adversarial time-varying filtering to user audio in real-time. VoiceBlock modifies speech by specifying the frequency magnitude response of a standard finite impulse response (FIR) filter. By carefully varying the filter's response many times per second, the speech is de-identified to an automated system while speaker identity is preserved for human listeners. This is accomplished without introducing noisy artifacts at the waveform, as filtering can only amplify or attenuate frequency energy present in the original signal through multiplication in the Fourier domain (see Section 3.3 for details of the filtering implementation used in our experiments).

VoiceBlock consists of three main modules, shown in Figure 1: (1) an **encoder** module extracts acoustic features from input audio frames, (2) a recurrent **bottleneck** module incorporates context from past frames to predict a set of adversarial filter controls for each frame, and (3) a **decoder** module regularizes the predicted filter controls and applies them to the corresponding frame of input audio. We provide an overview of each component below, and further details in Appendix A.

### 3.1 Encoder

The encoder module extracts acoustic features from 16kHz audio (double the sample rate of telephone speech) segmented into frames of 256 samples with 50% overlap. Though our model can operate in fully causal fashion, we find we achieve stronger attacks using a lookahead of five frames (see Section 3.2), resulting in a theoretical latency of 56ms. For reference, Verizon reports 25ms or less is typical latency for voice-over-IP communication in North America and Europe [56], while experimental data [26] show that 130ms of latency is acceptable for high-quality voice communications. Therefore, adding VoiceBlock to the signal chain leaves latency well within this limit.

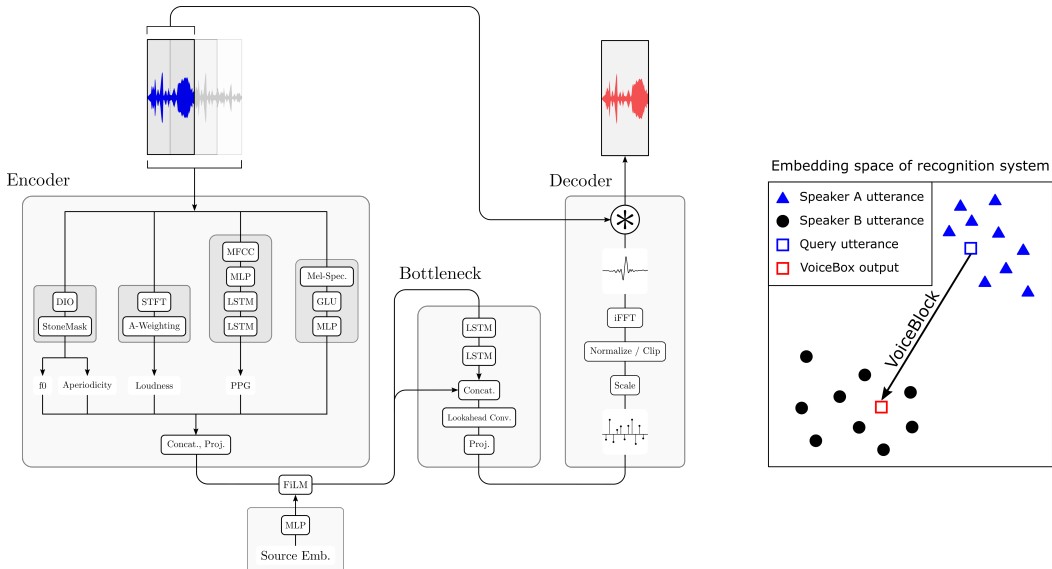

Figure 1: **Left**: The proposed VoiceBlock architecture. Acoustic features extracted by the encoder are fed to the recurrent bottleneck to predict filtering controls, which are regularized and applied to the input by the decoder to obtain adversarial audio. **Right**: VoiceBlock adversarially perturbs the user's audio stream such that any extracted queries are scored by the system as dissimilar to the user's enrolled utterances, hampering identification or retrieval.

Audio frames entering the encoder are passed to four sub-modules in parallel to obtain the following features.

**Pitch features**: Given the known sensitivity of speaker-recognition models to pitch [3] and the importance of spectral structure in delineating linguistic content, we extract fundamental frequency and aperiodicity estimates for each frame using the DIO algorithm [36]. Pitch estimates are refined by the StoneMask algorithm [37]. For both stages, we use the `pyworld` [22] implementation of the WORLD vocoder [37] (MIT license).

**Loudness**: We take an A-weighted average of each frame's log-magnitude spectrum to obtain a loudness estimate [35].

**Phonetic posteriorgrams**: Following the method of Ronssin & Cernak [48], we use a trained phoneme classifier with frozen weights to encode linguistic content. The classifier consists of a multi-layer perceptron followed by two LSTM [21] layers and a linear classification layer, and takes as input 13 mel-frequency cepstral coefficients (MFCC) with first- and second-order deltas for each frame. We train our phoneme classifier on the "train-clean-100" subset of the LibriSpeech dataset [42] using frame-aligned phoneme labels [33, 34] (CC-BY 4.0 license). Rather than directly use the classifier's predicted distributions over phoneme labels – known as phonetic posteriorgrams (PPGs) [19] – we discard the classification layer after training and pass along the output of the final LSTM layer, which Ronssin & Cernak also refer to as PPGs in their architecture.

**Spectrogram features**: Finally, we use a simple network consisting of a gated linear unit and multi-layer perceptron operating independently on each mel-spectrogram frame—a widely used speech representation [25]—to capture any residual information. We note that Engel et al. [14] also make use of a spectrogram encoder, alongside pitch and loudness features, to control differentiable signal-processing components.

For each frame, the above features are concatenated and projected linearly to obtain a low-dimensional encoding. We then introduce additional speaker information by passing a fixed, pre-computed embedding of the source speaker through a FiLM layer [45] to modulate the encoder output. This embedding helps to guide the de-identification task, allowing us to train a single model capable of de-identifying arbitrary users. This imposes a requirement that users must record a small amount of speech to obtain an embedding before first using VoiceBlock. We find that less than a minute of

speech is sufficient, and use a pre-trained ResNetSE34V2 model [20] to compute embeddings (see sections 4.1, 4.2).

## 3.2 Bottleneck

Encodings entering the bottleneck are passed through two LSTM layers, and the outputs are concatenated with a skip connection from the encoder. To enable streaming, we use unidirectional LSTM layers and pass concatenated outputs through a small lookahead convolutional network [57] to incorporate information from future frames at the expense of a small fixed delay. We find a lookahead of 5 frames (48 ms) is sufficient to craft strong de-identification attacks. Finally, a linear projection layer maps the concatenated representations to a vector of filter controls, with each element representing the unnormalized frequency magnitude response of a filter band for the current frame.

## 3.3 Decoder

Our decoder applies time-varying filtering to the input audio based on frame-wise controls obtained from the bottleneck. For our task, we find that a filtering-based decoder affords a number of advantages over the synthesis-based (e.g. transposed-convolutional) decoder architectures present in many audio-to-audio models [13]:

- The filtering module is not prone to the periodic upsampling artifacts that transposed-convolutional architectures often introduce [46].

- Our decoder is a simple deterministic module with no trainable parameters, keeping Voice-Block lightweight.

- The capacity of the decoder and conspicuousness of perturbations can easily be constrained in terms of the number of filter bands or their allowed range of motion, providing interpretable control to the user.

To regularize and apply filter controls to each frame of audio, we use a method similar to that of O'Reilly et al. and Engel et al. [41, 14]. We apply sigmoid scaling to bound filter controls to the range $[0, 2]$ and clip deviations from unity beyond a fixed value $\epsilon$. Each set of scaled filter controls is transformed into a time-domain impulse response via the inverse Fourier transform. We shift the impulse response to zero-phase (symmetric) form, apply a Hann window, and finally convolve with the corresponding input audio frame by taking the Fourier transform and performing element-wise multiplication. After compensating for shift, we overlap-add the resulting frames with a Hann window to obtain the final filtered audio. We discuss the details of our buffered streaming implementation in Appendix A.3. Our VoiceBlock model is implemented in PyTorch [2] and contains 6.3m trainable parameters, and 7.5m in total, counting the frozen phoneme encoder.

## 3.4 Training objective

To demonstrate the ability of VoiceBlock to perform inconspicuous privacy-preserving audio transformations, we train our model to attack speaker recognition systems. Given a large database of speech recordings and access to a user's audio stream, a surveilling entity may seek to (a) identify the user by matching their speech against recordings of known provenance in the database, or (b) retrieve other utterances of the user from the database. Systems designed for these tasks – speaker recognition and retrieval, respectively – often rely on neural network models to map speech utterances to a low-dimensional embedding space in which distance corresponds to speaker similarity [15]. In this work, we consider models $f$ for which the speaker distance $D_f$ between utterances $u$ and $v$ can be measured via a cosine distance between embeddings $f(u)$ and $f(v)$:

$$D_f(u, v) = 1 - \frac{f(u) \cdot f(v)}{\|f(u)\|_2 \|f(v)\|_2} \tag{1}$$

At inference time, *query* audio from the user is embedded and scored for similarity against *enrolled* utterances – pre-computed embeddings stored in the database. Scores may be evaluated for all enrolled embeddings, or for only a representative of each unique speaker (e.g. speaker centroids). The highest-ranking result or results (e.g. the closest speaker identity) are then returned. We do not distinguish between recognition and retrieval tasks, and refer to both under the umbrella of "speaker

recognition." This is because from a privacy standpoint, the objective in each task is identical: alter the query audio to prevent valid matches, thereby de-identifying the user.

A variety of methods have been proposed to efficiently compute and search over low-dimensional speaker representations. Generally, a hashing algorithm is applied to speaker embedding vectors to reduce storage and search costs [50, 30, 15]. Because the resulting hash representation merely serves as an efficient point of access for embedding-space distances, we omit hashing algorithms from consideration and define our attack objectives on the embedding space directly.

Given a speaker embedding model, we aim to modify query audio drawn from a user's stream such that its embedding-space distance from any enrolled utterances of the user is large, and its evaluated similarity is small. We quantify this de-identification in terms of distance thresholds in the embedding space, set according to percentiles of the estimated distribution of all inter-speaker embedding distances. Let $P_r$ represent the distance corresponding to the $r^{\text{th}}$ percentile of this distribution; then for query audio $u$ and enrolled utterance $v$, $D_f(u, v) > P_r$ implies that roughly $r$ percent of database entries should be scored as more similar to $u$ than $v$. Thus, we can set distance thresholds that correspond directly to the strength of de-identification applied to user audio, and construct a loss function that penalizes our model when the embedding-space cosine distance between query and enrolled utterances falls below the threshold. To do so, we use a variant of the adversarial loss proposed by Zhang et al. [62]. Let $f$ represent the victim model, $g$ our VoiceBlock network, $u$ an utterance from our user's audio stream, and $P_r$ our de-identification threshold; then

$$\mathcal{L}_{adv}(f, g, u) = (P_r - D_f(g(u), u) + \kappa)^+ \tag{2}$$

where $(\cdot)^+ = \max(\cdot, 0)$ and $\kappa$ is a confidence parameter encouraging the attack to fully cross the threshold. To ensure that our VoiceBlock model learns to perturb user audio inconspicuously, we incorporate an additional loss function to penalize perceptible filtering artifacts. We compute the combined waveform $L_1$ and multi-resolution spectrogram losses proposed by Defosséz et al. [13] on the clean and adversarial audio:

$$\mathcal{L}_{aux}(g, u) = \mathcal{L}_{stft}(u, g(u)) + ||u - g(u)||_1 \tag{3}$$

where $\mathcal{L}_{stft}$ is given by a sum of magnitude and spectral convergence losses computed over several spectrogram resolutions. We provide further details in Appendix A.2. Combining the above adversarial and auxiliary losses, we obtain our final attack objective:

$$\mathcal{L} = \mathcal{L}_{adv} + \mathcal{L}_{aux} \tag{4}$$

## 4 Experimental design

We describe experiments used to validate the claimed advances of our work, namely that VoiceBlock can de-identify speech in real-time while remaining significantly less conspicuous than existing methods of similar effectiveness. We first introduce the models, datasets, and attacks considered in our experiments. Following this, we detail our experiment configurations and present the results of both objective and subjective evaluations.

### 4.1 Speaker recognition models

**ResNetSE34v2:** We train attacks against the ResNetSE34v2 model [20] provided in the `VoxCeleb Trainer` repository [12] (MIT License). The model takes mel-spectrogram inputs and uses 2D convolutions with residual connections, squeeze-and-excitation, and attentive statistics pooling to generate frame-level features and aggregate them into 512-dimensional speaker embeddings.

**Y-Vector:** To examine the transferability of our approach against unseen systems, we evaluate trained attacks against the Y-Vector model proposed by Zhu et al. [63]. The model uses a multiscale 1D-convolutional waveform encoder to extract acoustic features, followed by squeeze-and-excitation blocks, feature aggregation, and a time-delayed neural network to map variable-length utterances to 128-dimensional speaker embeddings.

Both the ResNetSE34v2 and Y-Vector models were trained on the development set of the VoxCeleb2 dataset [11]. Note that while our attack is causal—modulo a short, five-frame lookahead—we do not impose the same restriction on a hypothetical surveillance system; instead, we evaluate our attack against strong, non-causal systems capable of aggregating speaker characteristics from across full utterances before rendering predictions.

### 4.2 Datasets

**LibriSpeech:** (CC-BY 4.0) We use both the `train-clean-100` and `test-clean` subsets of the LibriSpeech dataset [42] for training VoiceBlock. The former comprises 28,539 utterances from 251 speakers while the latter comprises 2,620 utterances from 40 speakers.

**VoxCeleb1:** (CC-BY 4.0) To simulate large-scale surveiling speaker recognition, we evaluate attacks on the VoxCeleb1 dataset [39], comprising 153,516 utterances from 1,251 speakers. This also ensures that no evaluation speakers are seen during training. As with the LibriSpeech dataset, we trim or pad all utterances to 4 seconds.

For all experiments, we carefully divide the data to imitate a realistic attack setting. During training, we select fifteen utterances (one minute total) from each source speaker in the training set and compute embeddings using the ResNetSE34v2 (MIT License) model. The centroid of these embeddings is then used as an enrolled target for the computation of the adversarial loss with all utterances of that speaker. We find that this produces stronger attacks than using individual utterance embeddings as targets, possibly by ensuring more consistent gradient information across the optimization. For our VoiceBlock attack (see Section 4.3), we select a further ten utterances (40s total) from each source speaker in the training set and again compute embeddings using the ResNetSE34v2 model. The centroid of these embeddings is then fed as a fixed conditioning vector to the VoiceBlock model alongside all utterances from that speaker (see Section 3.1). Similar to training, during evaluation we select fifteen utterances per speaker as a query set. We again select a further ten utterances to serve as conditioning for the VoiceBlock attack. Finally, twenty utterances of each speaker are enrolled in the speaker recognition system, serving as the database against which query utterances are matched.

### 4.3 Attack algorithms

We perform untargeted attacks using the following algorithms. **VoiceBlock:** We implement the proposed VoiceBlock attack as described in Section 3 and Appendix A and train for 10 epochs on 3 NVidia RTX 2080 Ti GPUs; this takes approximately 40 minutes. **Universal:** We optimize a short (2s) universal additive perturbation for 10 epochs using the established penalty method [32, 62, 27] and the same adversarial objective as VoiceBlock. On 3 NVidia RTX 2080 Ti GPUs, this takes approximately 16 minutes. To limit the perceptibility of the attack, we scale the perturbation to have $L_\infty$ norm 0.08 times that of the unperturbed speech. During training and evaluation, the perturbation is aligned arbitrarily with query utterances and looped to match durations, serving as a constant adversarial "background" signal. **White noise:** We add Gaussian noise to utterances at the waveform representation at a signal-to-noise ratio of $-10$dB. **Spectral gating:** We modify the "Kenansville" attack of Abdullah et al. [1] to allow for streaming use by performing spectral gating at all frames, using a threshold of 4dB relative to the maximum-energy spectral bin of each frame.

### 4.4 Objective evaluation

We evaluate attacks in a large-scale closed-set speaker recognition task. First, we use the `test-clean` LibriSpeech subset to obtain a rough estimate of the distribution of distances between an individual utterance and the centroids of each distinct speaker in the embedding space of the ResNetSE34V2 model. To encourage strong de-identification attacks, we take the distance corresponding to the 25^th percentile of this distribution as the target threshold $P_{25}$ for our training loss (see Section 3.4). We train each attack as discussed above.

We evaluate the closed-set recognition of all attacks over the VoxCeleb1 dataset. We compute the distance between each query embedding and the centroid of all embeddings of each speaker; recogntion is then performed by returning the speaker identity of the nearest speaker centroid. We find this is slightly more accurate and robust to attack than using the identity of the nearest embedded utterance. We perform exact nearest-neighbors search over the embedding space, and report the top-1 (T-1) and top-10 (T-10) accuracy of the relevant speaker recognition model given both clean and adversarial queries. Additionally, we compute the following objective speech quality metrics over the clean and adversarial query audio as a proxy measure of the imperceptibility of attacks: Perceptual Evaluation of Speech Quality (PESQ) [47], operating in the wide-band configuration and using the `python-pesq` implementation [58] (MIT license); and Short-Time Objective Intelligibility (STOI) [54], using the `pystoi` implementation [43] (MIT license).The results of our evaluation are presented in table 1.

Table 1: Results of our objective evaluation. We perform attacks on the VoxCeleb1 dataset against seen (ResNetSe34v2) and unseen (Y-Vector) speaker recognition models, and compute both top-1 and top-10 recognition accuracies. Additionally, we compute a set of objective speech quality metrics for each attack.

| Approach | Speech Quality Metrics | | ResNetSe34V2 | | Y-Vector | |
|---|---|---|---|---|---|---|
| | PESQ ↑ | STOI ↑ | T-1↓ | T-10↓ | T-1↓ | T-10↓ |
| White noise | 1.03 | 0.41 | 0.16 | 0.47 | 0.00 | 0.01 |
| Spectral gating | 1.11 | 0.63 | 0.02 | 0.12 | 0.03 | 0.13 |
| Universal | 1.35 | 0.78 | 0.09 | 0.20 | 0.48 | 0.73 |
| VoiceBlock | 3.74 | 0.92 | 0.03 | 0.10 | 0.35 | 0.66 |
| No attack | 4.64 | 0.99 | 0.99 | 0.99 | 0.96 | 0.99 |

We evaluate the real-time performance of the streaming implementation of the VoiceBlock attack described in Appendix A.3 by measuring its average real-time factor (RTF). We measure performance on a single thread on two different CPUs, an Intel i7-5600U @ 3.2 GHz and an Apple M1 Chip. In the streaming configuration, VoiceBlock processes a chunk of 4 overlapping frames at a time, equivalent to 640 samples / 40 ms. We compute the average RTF for processing a chunk over all chunks in a 4 second audio clip, and find that VoiceBlock has an average RTF of .255 on the Intel i7-5600U and .200 on the Apple M1.

## 4.5 Subjective evaluation

For our subjective evaluation, we sampled 100 clean speech recordings from the VoxCeleb1 dataset. To standardize comparisons, we trimmed or padded all utterances to 4 seconds. Each of the four attacks described in Section 4.3 was applied to all 100 recordings, resulting in 100 comparison sets with five recordings per set: the original clean speech and the speech modified by each of the four attacks. These sets were then evaluated in a MUSHRA-style listening study [9] deployed on Amazon Mechanical Turk using the open-source, MIT-licensed Reproducible Subjective Evaluation (ReSEval) [38] system. IRB approval was obtained prior to conducting this study, and there were no known risks to the participants in this study.

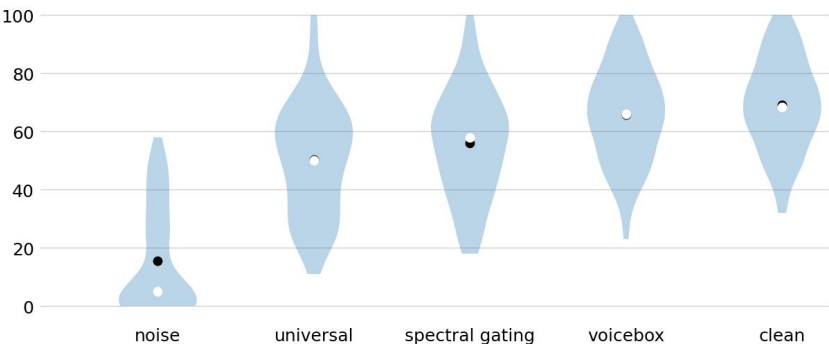

Figure 2: Distributions of quality ratings from our crowdsourced subjective listening MUSHRA-style test on audio quality. Higher numbers are better. Black dots are means and white dots are medians. Wilcoxon signed-ranked tests between all pairs of conditions show statistical significance at $p < 0.05$.

We recruited 20 participants. Participants were screened with a listening test prior to beginning the study; those that passed rated 20 comparison sets. For each comparison set, the participant was asked to listen to and rate the relative audio quality of each of the five audio files on a scale from 0 to 100. We omitted responses by four participants who failed our prescreening listening test and nine who rated the white noise attack (our low anchor) as superior in quality to ground-truth clean audio (our high anchor), giving us a total of 140 five-way comparisons. For more details of our crowdsourced subjective evaluation, see Appendix C.

The results of our crowdsourced subjective evaluation can be found in Figure 2. We find VoiceBlock is preferred to all other methods, and exhibits similar perceptual quality to ground-truth speech recordings. The difference between VoiceBlock and ground-truth audio is significant ($p < 0.05$) using a Wilcoxon signed-rank test ($p = 0.024$) but not significant using Welch's T-test ($p = 0.071$), which assumes that human perceptual scores are normally distributed but with potentially different variances. All other pairs of conditions are significantly different using either test.

### 4.6   Additional experiments

In Appendix B.1, we conduct ablation studies on the VoiceBlock encoder configuration, lookahead length, and choice of auxiliary loss. We also conduct supplementary experiments demonstrating the robustness of VoiceBlock against a deep network-based speech enhancement model, examining its performance under the assumption that adversarial queries are enrolled by the surveilling speaker recognition system, and evaluating the intelligibility of processed audio using an automatic system. These experiments are detailed in appendices B.2, B.3, and B.4 respectively.

## 5   Ethics

The main goal of our work is to show that large-scale, untargeted automated surveillance can be hindered by introducing inconspicuous perturbations to audio in real-time. Because VoiceBlock does not conceal speaker identity from human listeners, it still allows high-effort targeted surveillance (e.g. authorized human-attended wiretaps of criminal enterprises). In this way, we hope to return to the status quo of the 20th and early 21st centuries – in which the need for human listeners provided an important check on mass surveillance.

Those wishing to avoid all voice identification by human or machine may use existing voice conversion technologies. Voice conversion is not, however, appropriate in many contexts where some measure of privacy from mass surveillance may be desired. Consider telehealth: a medical expert or therapist discussing sensitive topics with a patient may gain great insight from hearing the nuance of the patient's voice. In this setting, methods that alter one's voice beyond human recognition (e.g. voice conversion) would necessarily degrade the interaction.

The approach of VoiceBlock could also, in concept, be used to fraudulently access information or services protected by speaker verification. However, VoiceBlock performs perceptually inconspicuous filtering and the speech produced would still sound like the original speaker. As such, it could not pass human inspection. Both voice-conversion and speech synthesis-based attacks are much more suited to this purpose, as both produce speech in a voice perceptually similar to the target speaker.

Our approach can be thought of as leveraging the asymmetry between system and attacker attention inherent to mass surveillance, using inconspicuous perturbations to evade large-scale systems that must render predictions in bulk. This same asymmetry is not necessarily present when trying to bypass authentication mechanisms and access tightly-guarded services.

## 6   Conclusions

Our experimental results indicate VoiceBlock can de-identify speech from arbitrary unseen users in real-time on a standard M1 CPU. The de-identified speech is of significantly higher audio quality than competing methods, as reported by a subjective listener study and as measured through standard metrics of speech intelligibility. Our method also achieves nontrivial de-identification results against a system it was not trained on, outperforming a far more perceptible universal attack. This is notable given that we take no explicit steps to improve the transferability of our method, which leverages a lightweight model trained on a small corpus of clean speech without augmentation. While pure signal-processing approaches such as spectral gating and white noise transfer between systems more successfully, they severely degrade audio quality, resulting in much lower listener quality ratings. This limits their practicality in real-world voice communications. By contrast, our method produces adversarial audio that is virtually indistinguishable from the clean source, as indicated by our subjective evaluation.

We view our method as an initial step towards protecting users from indiscriminate mass surveillance. A number of obvious directions for future work stand out, such as improving the transferability of the

adversarial examples crafted by our system [40], and evaluating attacks against real-world speaker recognition systems and over real-world communication channels. We hope this work encourages further exploration of the applications of audio-to-audio models for protecting user privacy.

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
