# A   VoiceBlock implementation

## A.1   Architecture

We provide additional details of the VoiceBlock architecture used in our experiments.

**Phoneme encoder:**   The phoneme encoder consists of a feedforward network with two hidden layers of size 256 and ReLU activations, followed by two LSTM layers with hidden size 256. Using the LibriSpeech "train-clean-100" dataset and aligned phoneme labels provided by Lugosch et al. [33], we train the phoneme encoder with cross entropy loss for 25 epochs using the Adam optimizer with learning rate 1e-3. During training, a linear layer is appended to the phoneme encoder to render predictions over phoneme labels; we discard this layer after training.

**Spectrogram encoder:**   We pass 64-bin mel-spectrograms through a $1 \times 1$ convolutional layer and gated linear unit (GLU) operating along the frequency axis, followed by a two linear layers with leaky ReLU activations and hidden size 512. The convolution layer upsamples the channel (frequency) dimension to 1024, from which the GLU obtains a 512-dimensional representation to pass to the linear layers.

**Source speaker conditioning:**   For each frame of audio, we concatenate the 256-dimensional output of the phoneme encoder, the 512-dimensional output of the spectrogram encoder, and the one-dimensional pitch / periodicity / loudness features and linearly project to obtain a 512-dimensional vector. A fixed 512-dimensional source speaker embedding obtained from the ResNetSE34V2 model is passed through a 2 linear layers with leaky ReLU activations and hidden size 512, followed by batch normalization and a final linear layer to obtain a 1024-dimensional vector holding scales and biases for a feature-wise affine transformation (FiLM) of the encoder output. The affine transformation is applied and the resulting vector is passed to the bottleneck.

**Bottleneck:**   Our bottleneck consists of two LSTM layers with hidden size 512. The encoder output is passed through the LSTM layers with a skip connection concatenated at the output. Following the application of a lookahead convolutional network, the resulting vector is projected linearly to obtain a 128-dimensional vector of filter controls for the current frame.

**Decoder:**   We perform filtering with 128 bands per frame, and set $L_\infty$ bound $\epsilon = 0.5$ to further constrain deviations from a "neutral" control configuration of $\vec{1}$. We experimented with alternative methods of regularizing predicted filter controls, such as $L_2$ normalization, but found that simple $L_\infty$ clipping yielded satisfactory results.

**Training:**   We train both VoiceBlock and the universal baseline attack using the Adam [29] optimizer with learning rate $1\mathrm{e}{-4}$ for 10 epochs over the LibriSpeech "train-clean-100" subset.

## A.2   Auxiliary loss

We briefly summarize the multi-resolution STFT loss of Defosséz et al. [13] used in the training of our attacks. The loss is given by

$$\mathcal{L}_{aux}(g, u) = \mathcal{L}_{stft}(u, g(u)) + ||u - g(u)||_1 \tag{5}$$

where $\mathcal{L}_{stft}$ is given by a sum of magnitude and spectral convergence losses computed over $M$ spectrogram resolutions:

$$\mathcal{L}_{stft}(a, b) = \sum_{i=1}^{M} \left[ \mathcal{L}_{sc}^{(i)}(a, b) + \mathcal{L}_{mag}^{(i)}(a, b) \right] \tag{6}$$

$$\mathcal{L}_{sc}(a, b) = \frac{||\ |STFT(a)| - |STFT(b)|\ ||_F}{||\ |STFT(a)|\ ||_F} \tag{7}$$

$$\mathcal{L}_{mag}(a, b) = ||\ \log |STFT(a)| - \log |STFT(b)|\ ||_1$$

$$\tag{8}$$

Here $|| \cdot ||_F$ and $|| \cdot ||_1$ refer to the Frobenius and $L_1$ norms, respectively. Following Defosséz et al., we use $M = 3$ spectrogram resolutions with hop sizes of 50, 120, and 240 samples, FFT lengths of 512, 1024, and 2048, and window lengths of 240, 600, and 1200, respectively.

### A.3 Streamer

We implement a version of VoiceBlock for processing audio streams in real time. All components are modified to support the computation of perturbations on overlapping windows of audio while the input and output streams process non-overlapping chunks. Where possible, components are complied to TorchScript to improve performance. Three buffers are added to process a stream of audio into overlapping chunks:

- Last Frame Input Buffer: Contains the last 128 samples of the last input audio to be appended to the oncoming chunk, before computing the overlapping windows.
- Lookahead Buffer: Contains overlapping frames to be used as lookahead.
- Output Buffer: Contains the last 128 samples output by VoiceBlock, to be overlap-added with the next processed chunk.

Code for our streaming implementation can be found at `https://interactiveaudiolab.github.io/project/voiceblock.html`.

## B  Additional experiments

### B.1  Ablation study

To better understand the contribution of various aspects of the proposed model architecture and training procedure, we perform the following ablations. **Encoder:** We examine the effectiveness of models using a subset of the encoder components discussed in Section 3.1. Results are presented in Table 2. **Lookahead:** we vary the number of lookahead frames passed to the model during training and inference. At a lookahead of zero, the model operates in a causal manner. We report model effectiveness and theoretical latency as a function of lookahead in Table 3. **Auxiliary loss**: we replace the combined waveform and multi-resolution spectrogram loss of Defosséz et al. [13] with a waveform-only $L_1$ loss, the mel-frequency cepstral coefficient cosine-similarity loss used by O'Reilly et al. [41], and an ASR feature-matching loss. Results are reported in table 4. Note that the latter model variant outperforms the "main" VoiceBlock model presented in this work along all objective metrics, although we were unable to conduct a subjective evaluation to verify apparent gains in perceptual quality. For all ablations, we perform attacks on the ResNetSE34 model using the experimental configuration detailed in Section 4.

### B.2  Attack robustness to preprocessing

In real-world settings, user audio may travel through various preprocessing stages before reaching a speaker recognition model. To simulate the presence of such stages, we perform attacks using the experiment configuration discussed in Section 4.3 but pass query utterances through a pretrained

Table 2: Results of ablations on the encoder module. The full VoiceBlock encoder consists of a spectrogram network ("Spec"); a phoneme predictor network ("PPG"); pitch, aperiodicity, and A-weighted loudness features ("DSP"); and speaker-conditioning via the embeddings of a pretrained recognition model ("condition")

| VoiceBlock Encoder | Speech Quality Metrics | | ResNetSe34V2 | |
| | PESQ ↑ | STOI ↑ | T-1↓ | T-10↓ |
|---|---|---|---|---|
| Spec | 3.41 | 0.89 | 0.02 | 0.10 |
| · Spec + PPG | 3.56 | 0.89 | 0.03 | 0.14 |
| Spec + PPG + DSP | 3.64 | 0.90 | 0.02 | 0.10 |
| Spec + DSP + Condition | 3.67 | 0.91 | 0.02 | 0.11 |
| Spec + PPG + DSP + Condition | 3.74 | 0.92 | 0.03 | 0.10 |

Table 3: Results of varying the lookahead length in frames. We train and evaluate models at matched lookahead lengths. For each lookahead, we note the minimum theoretical latency required in milliseconds.

| Lookahead | Minimum Latency | Speech Quality Metrics | | ResNetSe34V2 | |
| | ms | PESQ ↑ | STOI ↑ | T-1↓ | T-10↓ |
|---|---|---|---|---|---|
| 0 | 16 | 3.91 | 0.93 | 0.04 | 0.12 |
| 1 | 24 | 3.93 | 0.93 | 0.03 | 0.13 |
| 2 | 32 | 3.83 | 0.92 | 0.03 | 0.15 |
| 5 | 56 | 3.74 | 0.92 | 0.03 | 0.10 |

Demucs [13] speech enhancement model en route to the ResNetSE34v2 recognition system; results are reported in table 5. The speaker recognition system maintains its top-1 and top-10 accuracy on clean queries. In contrast to Chiquier et al. [10] we do not incorporate the enhancement model into our adversarial optimization, meaning that Demucs essentially functions as an unseen adversarial defense. We find that VoiceBlock loses almost no effectiveness when queries are passed through Demucs; by comparison, the universal attack loses most of its effectiveness, as Demucs is able to separate the low-magnitue but noisy perturbation from clean speech. Both the white noise and spectral gating attacks retain their effectiveness, presumably because they degrade audio beyond Demucs' capability to provide coherent reconstructions of the original speech. We leave as future work the exploration of the robustness of VoiceBlock against more sophisticated preprocessing pipelines and adversarial defenses.

Table 4: Results of training VoiceBlock with various auxiliary loss functions. We consider a simple waveform regression loss ("$L_1$") and the mel-cepstral cosine-similarity loss used by O'Reilly et al. ("MFCC-cosine") [41] as simple drop-in replacements for the combined waveform/spectrogram loss of Defosséz et al. [13] ("$L_1$ + MRS"). Additionally, we consider a feature-matching loss on the acoustic representations produced by a Wav2Vec 2.0 [5] automatic speech recognition model ("ASR"). This loss is computed by taking the $L_1$ distance between acoustic feature vectors produced by the Wav2Vec 2.0 encoder over clean and adversarial utterances, scaled by a factor of $1e^{-6}$.

| Auxiliary Loss | Speech Quality Metrics | | ResNetSe34V2 | |
| | PESQ ↑ | STOI ↑ | T-1↓ | T-10↓ |
|---|---|---|---|---|
| $L_1$ | 3.49 | 0.89 | 0.02 | 0.07 |
| MFCC-cosine | 3.34 | 0.89 | 0.02 | 0.07 |
| ASR | 4.08 | 0.94 | 0.02 | 0.09 |
| $L_1$ + MRS | 3.74 | 0.92 | 0.03 | 0.10 |

Table 5: Top-1 (T-1) and top-10 (T-10) recognition accuracy of the ResNetSE34v2 system on the de-identification attacks described in Section 4.3 when all query audio is passed through a Demucs [13] speech enhancement preprocessing stage

|  | **ResNetSe34V2** | | **+Demucs** | |
| **Approach** | **T-1 ↓** | **T-10 ↓** | **T-1 ↓** | **T-10 ↓** |
| --- | --- | --- | --- | --- |
| White noise | 0.13 | 0.40 | 0.02 | 0.09 |
| Spectral gating | 0.02 | 0.11 | 0.02 | 0.11 |
| Universal | 0.14 | 0.22 | 0.87 | 0.97 |
| VoiceBlock | 0.02 | 0.10 | 0.04 | 0.15 |
| No attack | 0.97 | 0.99 | 0.96 | 0.99 |

Table 6: For sets of 15 clean and adversarial query utterances, we compare the top-1 (T-1) and top-10 (T-10) accuracy of speaker recognition with the ResNetSE34v2 model under three conditions. **Clean profile**: 20 clean utterances are enrolled per speaker. **Mixed profile**: 10 clean and 10 adversarial (VoiceBlock) utterances are enrolled per speaker. **Adversarial profile**: 20 adversarial (VoiceBlock) utterances are enrolled per speaker.

|  | **Clean profile** | | **Mixed profile** | | **Adversarial profile** | |
| **Query processing** | **T-1 ↓** | **T-10 ↓** | **T-1 ↓** | **T-10↓** | **T-1 ↓** | **T-10 ↓** |
| --- | --- | --- | --- | --- | --- | --- |
| VoiceBlock | 0.02 | 0.10 | 0.51 | 0.80 | 0.78 | 0.93 |
| None | 0.97 | 0.99 | 0.83 | 0.95 | 0.09 | 0.28 |

## B.3 Enrollment of adversarial queries

It is possible that adversarially-perturbed query utterances extracted from a VoiceBlock user's audio stream may themselves be enrolled by a surveiling speaker recognition system. In such cases, there are two possibilities:

1. VoiceBlock successfully de-identifies user speech, and adversarial queries are enrolled as a separate speaker profile from any existing clean utterances of the user

2. VoiceBlock fails to de-identify user speech, and adversarial queries are incorporated into an existing profile of the user containing clean utterances.

We examine both scenarios, again using the VoiceBlock attack discussed in Section 4.3. Results are presented in table 6. We find that while VoiceBlock is highly effective at de-identifying users against a profile constructed from clean (unperturbed) utterances, its de-identification performance suffers significantly under both of the aforementioned conditions. This suggests that additional work is required to ensure that a VoiceBlock user remains a "moving target" to surveilling speaker recognition systems. One possible solution is to leverage targeted rather than untargeted attacks: by proactively "spoofing" specific (random) locations in the embedding space, VoiceBlock may hamper the creation of a single matching enrolled profile.

## B.4 Word error rate evaluation

To measure the intelligibility of speech processed by the proposed and baseline methods, we evaluate the word- and character-error rates of each attack described in Section 4.3 on the LibriSpeech `train-clean-360` subset using a Wav2Vec 2.0 automatic speech recognition model [5]. The `train-clean-360` dataset contains utterances and accompanying transcriptions from 921 speakers not seen during training. We do not use the VoxCeleb dataset because it does not have the necessary transcriptions. We partition each speaker's data into query, enrollment, and conditioning partitions, as described in Section 4.2. Our results are reported in Table 7.

Our results demonstrate that pure signal-processing attacks such as spectral gating and white noise result in word- and character-error rates 20+ times that of VoiceBlock. VoiceBlock also outperforms all methods except spectral gating on the de-ientification task, with which it performs competitively. This indicates our method's ability to preserve the intelligibility of speech while adversarially modifying speaker characteristics.

Table 7: De-identification performance of proposed and baseline models over the LibriSpeech `train-clean-360` dataset. To evaluate the intelligibility of the resulting audio, we ASR compute word- and character-error rates in addition to PESQ and STOI scores.

| | Speech Quality Metrics | | ResNetSe34V2 | | Wav2Vec 2.0 | |
| Approach | PESQ ↑ | STOI ↑ | T-1 ↓ | T-10 ↓ | WER ↓ | CER ↓ |
|---|---|---|---|---|---|---|
| White noise | 1.03 | 0.52 | 0.24 | 0.58 | 1.00 | 1.00 |
| Spectral gating | 1.13 | 0.64 | 0.03 | 0.18 | 0.50 | 0.29 |
| Universal | 1.38 | 0.86 | 0.09 | 0.24 | 0.16 | 0.08 |
| VoiceBlock | 3.72 | 0.94 | 0.05 | 0.22 | 0.02 | 0.01 |
| No attack | 4.64 | 0.99 | 0.99 | 0.99 | 0.01 | 0.01 |

## C Listening Study

The listening study consists of 20 evaluation tasks, where the participants are shown the following text, along with 5 four second long audio examples and 5 corresponding sliders with values from 0 to 100:

> Listen to all recordings of a person speaking. Then, move the sliders to **rate the quality of each audio file from 0 (worst) to 100 (best)**. The higher-quality audio files are the ones that are more natural sounding, or have fewer audio artifacts (e.g., clicks, pops, noise, or otherwise sound 'unnatural'). **Note** - Each slider cannot be moved until its corresponding audio file has been listened to in its entirety.

It takes approximately 15 minutes to complete all the evaluation tasks. Participants are also asked to complete a listening test before proceeding to the audio evaluation. The listening test involves the participant listening to two audio files, and reporting the number of tones they heard. For each audio file, the participants are given 3 tries to correctly report the number of tones played. If a participant succeeds in the listening test and completes the evaluation tasks, then they are paid 3.00 USD, or equivalently, 12.00 USD / hour. If the participant fails the listening test, then they are paid 0.50 USD.