# OpenReview forum: "VoiceBlock: Privacy through Real-Time Adversarial Attacks with Audio-to-Audio Models"
_NeurIPS.cc/2022/Conference — NeurIPS 2022 Accept_

### Official Review · Reviewer_G7js · 2022-07-03

**Rating:** 8
**Confidence:** 3
**Soundness:** 4 excellent
**Presentation:** 2 fair
**Contribution:** 3 good

**Summary:**

This paper presents VoiceBox, an adversarial audio attack which (1) dramatically lowers the accuracy of speaker recognition systems, (2) is mostly imperceptible to humans, and (3) can operate in real-time on live audio streams. The authors further demonstrate that their proposed model may also transfer to speaker recognition systems that it was not explicitly trained to fool.

**Questions:**

Can the authors could shed more light on this contradiction in the rebuttal, or otherwise rework their submission to address this concern in a more transparent fashion?

Other comments / questions:
- Especially in a paper largely concerning privacy, a section devoted to Ethics would be nice to see. While privacy is an admirable goal, might there be some potential ethical *downsides* of releasing a de-identification system like VoiceBox?
- Seems to be a known issue already, but the author should rework their results to include the ASR results which are only present in the online supplement
- Clarify code release? The checklist indicates that code is proprietary and will not be released, but the supplement implies that it will be. (My review gives the benefit of the doubt that the code will be released)


**Limitations:**

See above

**Strengths And Weaknesses:**

Overall, this is an interesting and well-written paper which proposes a nice method and demonstrates its efficacy through carefully-designed experiments and user studies. Moreover, the authors plan to release code and a reference implementation of the system which can run on commodity hardware. My main issue with this paper lies in a perceived contradiction in the core motivation (expanded below). However, despite this contradiction, I do think VoiceBox will be of interest to the voice privacy research community in both the short and long term, as well as possibly even to users in the short term.

**Contradiction in core motivation.** There is seemingly a large contradiction in the premise of this work: that in a landscape of automated surveillance via speaker recognition, we should work to design attacks that are imperceptible to humans, i.e., ones that _preserve the identity_ of the speaker to human listeners. I cannot understand why this is a desirable property. Suppose that, once VoiceBox is released, speaker recognition systems are augmented with a defense which detects if speech was modified by VoiceBox. In this case, users could face repercussions such as _denial of service_ (automatable) or _targeted retaliation_ (through manual audit to determine identity). In the worst case, if speaker recognition systems can also be trained to be  _robust_ to VoiceBox, users may even face _automated_ targeted retaliation, since both their use of VoiceBox and their identity would be known automatically.

**Why not voice conversion?** Given this contradiction, it seems to me that _voice conversion_ is a much more desirable goal, because it would at least remove the possibility of targeted retaliation against users (assuming that the voice conversion process eliminates the possibility of recovering original speaker identity). The authors mention voice conversion in related work, advocating in favor of their approach as it is “inconspicuous and minimally invasive to the user experience”. However it is historically unlikely that any attack which is “inconspicuous” to humans will be undetectable by a deep learning system, and it’s unclear what the “user experience” means in this context, so this justification is largely unsatisfying.

---

> ### Author Response · Authors · 2022-08-01
> **Response to reviewer G7js**
>
> We would like to thank the reviewer for their thoughtful reading of our manuscript and for providing detailed comments about how to improve the paper. Here, we address the reviewer’s comments and any corresponding revisions point-by-point:
>
> __1. "There is seemingly a large contradiction in the premise of this work: that in a landscape of automated surveillance via speaker recognition, we should work to design attacks that are imperceptible to humans, i.e., ones that preserve the identity of the speaker to human listeners. I cannot understand why this is a desirable property. …"__
>
> There are a number of communication contexts in which a user may wish to preserve the perceptual quality of their speech while gaining a measure of protection against invasive deep-learning systems. For example, a user may wish to use their natural speaking voice in a video chat with friends or in a livestream – all without the resulting audio data being parsed successfully by a speaker-recognition algorithm or used to train a voice clone. Tele-health is an example application where a measure of privacy may be desired, yet the practitioner (e.g. a therapist or medical expert) may gain value from hearing the nuance of the patient’s original voice in a perceptually-unaltered way. In such settings, voice conversion would necessarily degrade the experience of the user.
>
> __2. "Suppose that, once VoiceBox is released, speaker recognition systems are augmented with a defense which detects if speech was modified by VoiceBox. In this case, users could face repercussions such as denial of service (automatable) or targeted retaliation (through manual audit to determine identity)."__
>
> Automating denial of service for VoiceBox users reveals that surveillance is performed on a communication channel; this would likely cause VoiceBox users to switch to another communication channel, defeating the point of surveillance.
>
> From our perspective, forcing a manual audit is a positive outcome. The primary goal of our work is to demonstrate that the operation of large-scale, untargeted automated surveillance systems can be hindered by introducing inconspicuous perturbations in real-time. We view this as distinct from shielding individuals from high-effort targeted surveillance, which may in cases be employed to legitimate ends (e.g. authorized wiretaps of criminal enterprises). VoiceBox makes human-transparent modifications to stop automated voice recognition and forces a human to identify the voice. By thwarting the application of deep-learning systems to user audio data while leaving the original perceptual quality intact, we hope to return to the status quo of the 20th and early 21st centuries – in which effective surveillance of the content of voice communications required direct human intervention. We have clarified our discussion of this goal in a new Ethics section (see point 4 below).
>
> __3. "In the worst case, if speaker recognition systems can also be trained to be robust to VoiceBox, users may even face automated targeted retaliation, since both their use of VoiceBox and their identity would be known automatically? It seems to me that voice conversion is a much more desirable goal, because it would at least remove the possibility of targeted retaliation against users (assuming that the voice conversion process eliminates the possibility of recovering original speaker identity)."__
>
> We agree that the issue of detection is relevant and worth exploring; however, given that there exists a robust literature on voice-conversion detection systems, including challenges such as ASVSpoof [1], it is not clear a priori that the proposed system would fare worse than voice conversion systems in this regard. Regarding the possibility for high-effort, targeted recovery of speaker identity and retaliation, we refer to our discussion of point 2 above.
>
> We see the development of human-imperceptible approaches such as our own as orthogonal to the development of voice-conversion-based approaches, as both suit different needs. While voice conversion may grant a stronger guarantee of anonymity in many circumstances, it may hinder legitimate law enforcement (e.g. preventing the use of voice recordings as evidence) and degrade the quality of communication in situations where hearing the nuance of a person’s unaltered voice is essential (e.g. telehealth).

---

> > ### Author Response · Authors · 2022-08-01
> > **Response (continued)**
> >
> > __4. “Especially in a paper largely concerning privacy, a section devoted to Ethics would be nice to see. While privacy is an admirable goal, might there be some potential ethical downsides of releasing a de-identification system like VoiceBox?”__
> >
> > We agree with the reviewer’s recommendation, and have added an Ethics section (Section 5) in our revised manuscript.
> >
> > __5. “Seems to be a known issue already, but the author should rework their results to include the ASR results which are only present in the online supplement”__
> >
> > Appendix B.3 now holds the results of the ASR experiments presented in the online supplement.
> >
> > __6. “Clarify code release? The checklist indicates that code is proprietary and will not be released, but the supplement implies that it will be. (My review gives the benefit of the doubt that the code will be released)”__
> >
> > Our code, including a streaming implementation of the proposed model, is now available via the project webpage: https://master.d3hvhbnf7qxjtf.amplifyapp.com/. We will release our code publicly upon acceptance.
> >
> > We would like to reiterate our appreciation of the reviewer for their detailed comments, and we believe that the paper has been made stronger through our attempts to address them. Thank you!
> >
> > References
> >
> > [1] https://www.asvspoof.org/

---

> > > ### Comment · Reviewer_G7js · 2022-08-09
> > > **Thanks for the response**
> > >
> > > I really appreciate the detailed and thoughtful responses to my concerns! I had certainly not considered applications such as voice chat and telehealth, which are both great motivating examples for a system which maintains perceptual identity but impedes surveillance.
> > >
> > > The changes to the Appendix and the availability of code are also a welcome addition to this submission.
> > >
> > > I have increased my score to acknowledge these improvements to an already-strong paper.

---

### Official Review · Reviewer_LEcn · 2022-07-06

**Rating:** 7
**Confidence:** 4
**Soundness:** 3 good
**Presentation:** 3 good
**Contribution:** 3 good

**Summary:**

This manuscript proposes an adversarial attack for audio-to-audio model that preserves user’s privacy against speaker recognition models. The proposed approach is based on a seq2seq neural network model, trained with multi-task losses including an adversarial loss to attack the speaker recognition model, and an auxiliary loss for finite impulse response filtering and audio reconstruction.

**Questions:**

N/A

**Limitations:**

Satisfactory

**Strengths And Weaknesses:**

Strengths

* The proposed approach is interesting, and the technical are enough.
* The authors conducted both objective and subjective evaluations on the audios generated from the proposed methods, regarding the speech audio quality and the performance on (seen/unseen) victim speaker recognition models. Results show that the proposed approach significantly outperform baselines.
* The paper is well written and well organized. References are enough.
* Sample page and code are available.

Weaknesses
* The evaluations are only conducted on the generated audios. It would be more promising to have evaluations on a couple of real audio-to-audio tasks like enhancement, voice conversion, speech-to-speech translation. Although the generated audios have satisfactory quality from the evaluations on themselves, it is possible that the extracted features are very different since the proposed approach modulates their waveforms, causing regressions on these tasks

---

> ### Author Response · Authors · 2022-08-01
> **Response to reviewer LEcn**
>
> We would like to thank the reviewer for their thoughtful reading of our manuscript. Here, we address the reviewer’s comment:
>
> __“The evaluations are only conducted on the generated audios. It would be more promising to have evaluations on a couple of real audio-to-audio tasks like enhancement, voice conversion, speech-to-speech translation. Although the generated audios have satisfactory quality from the evaluations on themselves, it is possible that the extracted features are very different since the proposed approach modulates their waveforms, causing regressions on these tasks.”__
>
> We agree with the reviewer that there exist many tasks beyond speaker recognition in which deep models may be put to invasive or unethical use – including audio-to-audio tasks like voice conversion – and that our evaluations could be made more comprehensive by including such systems. However, in the paper, we deliberately maintain a narrow focus on the task of speaker recognition to allow space for a detailed discussion of the proposed approach. Other recent works have explored adversarial attacks on ASR [1] and voice conversion [2] from the perspective of user privacy, and we believe adapting our system to these and other tasks to be a strong direction for future work.
>
> The experiments detailed in Appendices B1 and B3 of our paper suggest that when trained only to evade speaker recognition, our system modifies audio in ways that do not significantly affect the performance of a Demucs speech-enhancement model or Wav2Vec 2.0 ASR model, respectively. Similarly, we would not necessarily expect our approach to impede the performance of models across other speech-processing tasks without modifying the training scheme to include an appropriate task-specific objective. We therefore leave such experiments to future work.
>
> We would like to reiterate our appreciation of the reviewer for their comments. Thank you!
>
> References
>
> [1] Mia Chiquier, Chengzhi Mao, and Carl Vondrick. Real-time neural voice camouflage. In
> International Conference on Learning Representations (ICLR), 2022.
>
> [2] Chien-yu Huang, Yist Y. Lin, Hung-yi Lee, and Lin-shan Lee. Defending Your Voice: Adversarial Attack on Voice Conversion. In SLT, 2021.

---

> > ### Comment · Reviewer_LEcn · 2022-08-08
> > **Thanks for the response**
> >
> > Thanks for the explanation of my questions! The newly added parts (from other reviewers' perspective) make the paper more solid. Ihave increased my score a little bit for this.

---

### Official Review · Reviewer_NLhW · 2022-07-12

**Rating:** 7
**Confidence:** 4
**Soundness:** 3 good
**Presentation:** 3 good
**Contribution:** 3 good

**Summary:**

This paper proposes a real-time audio conversation to de-identify speakers from a privacy perspective. One of the unique aspects of this model is to consider frequency domain filtering instead of noise addition, which seems to avoid the distortion of speech signals in terms of their speech quality measures (subjective and objective) while maintaining the speaker de-identification functions. This paper significantly contributes to the community by considering that the voice privacy issue has become a severe concern and audio applications require streaming/real-time operations.

**Questions:**

- Did you perform automatic speech recognition? I'm expecting that ASR would be robust against the "time-invariant" FIR filter, and I'm curious how it would degrade the performance due to the proposed method based on the time-variant FIR filter.
- Is it possible to evaluate this method with the Voiceprivacy challenge benchmark? Then, the paper can compare various techniques proposed in the challenge or literature.
- "VoiceBox" is the same name as the famous (Matlab) speech processing toolkit. https://scholar.google.com/citations?view_op=view_citation&hl=en&user=HEH9cboAAAAJ&citation_for_view=HEH9cboAAAAJ:u-x6o8ySG0sC. Although this is an old toolkit (based on Matlab!), I recommend the authors respect this prior work and avoid using this name.
- I would like to have more discussions on why the authors use the time-variant FIR filter with more scientific perspectives. I agree that a time-invariant FIR filter would be less distorted than the additive noise as it would just physically change the speaker or room characteristics. But I'm not very sure about the time-variant FIR filter.
- Can you discuss 56ms? I think this range would be accepted in many speech interface applications. For example, DNS challenges and other real-time speech processing tasks set a similar range for the latency constraint. However, some hearing devices set the latency constraint to less than 10 ms (you can check the Clarity challenge). This kind of discussion makes the paper more attractive since we can imagine the main application of this method.
- How much is the phoneme classifier part robust against the other languages, speaking styles, and noise environments? Librispeech "train-clean-100" is English/clean/read speech, which would not cover many speech variations.
- Can you discuss the RTF? Is 0.255 or 0.200 sufficiently light?


**Limitations:**

The paper describes the limitation, especially for real-world applications.

**Strengths And Weaknesses:**

Strengths
- Real-time speaker de-identification is a critical topic for voice privacy, and the motivation part of the paper clearly describes this importance.
- The paper carefully describes the reproducibility of the proposed method.
- Time-varying filter-based de-identification is novel compared with existing studies.
- The experimental results clearly demonstrate the improvement from the existing studies regarding both subjective and objective evaluation metrics.

Weaknesses
- The topic is specific to speech processing rather than generic machine learning and may not gain attention from general NeurIPS audiences. I think the paper would have a better fit for the speech conferences (e.g., Interspeech and ICASSP).
- The experimental evaluation should focus more on the latency. It would be good to have the performance vs. latency trade-off.
- Although the paper shows the overall improvement of the proposed method, the proposed method seems to be very different from the other methods, especially for the encoder part and training schemes in addition to the time-variant" FIR filter. For example, it is not clear about the experimental justification of this neural network architecture. The proposed method tries to disentangle various speech factors (pitch, phoneme, loudness) based on speech processing models in the encoder, but this architecture has no experimental validation. I suggest the authors provide the contribution of these factors (possibly through the ablation study of each aspect) to the overall performance.
- I have the same discussion for the loss function. There is no experimental justification for the proposed training scheme.

---

> ### Author Response · Authors · 2022-08-01
> **Response to reviewer NLhW**
>
> We would like to thank the reviewer for their thoughtful reading of our manuscript and for providing detailed comments about how to improve the paper. Here, we address the reviewer’s comments and any corresponding revisions point-by-point:
>
> __1. “The topic is specific to speech processing rather than generic machine learning and may not gain attention from general NeurIPS audiences. I think the paper would have a better fit for the speech conferences (e.g., Interspeech and ICASSP).”__
>
> We believe that preserving privacy in real-time digital communications is of broad interest, especially given the widespread adoption of deep learning technologies for audio analysis and synthesis. Similar work has been presented at general machine learning conferences (e.g. [1]). Moreover, despite our focus on audio, we hope that this work will spark discussion on how to protect other data modalities against misuse by deep learning systems in real-time (e.g. protecting streaming video in the case it is eventually passed to systems performing intrusive analysis or unauthorized synthesis).
>
> __2. “The experimental evaluation should focus more on the latency. It would be good to have the performance vs. latency trade-off.”__
>
> The authors of [8] and [9] indicate that end-to-end one-way latencies of up to 130-150ms are considered acceptable in voice-over-IP communications. If we take Verizon’s reported VoIP latencies [10] of 15ms in Europe and 25ms in North America as standard, then adding 56ms latency with our proposed system would appear to fall well under the acceptable limit, even accounting for additional latency introduced through the device or applications. We have revised the paper at line 135 to reflect this information.
>
> Given this, and the limited space available in the paper, we feel that it is more important to focus on other ablations. We hope the reviewer understands.
>
> __3. “Although the paper shows the overall improvement of the proposed method, the proposed method seems to be very different from the other methods, especially for the encoder part and training schemes in addition to the time-variant FIR filter. For example, it is not clear about the experimental justification of this neural network architecture. The proposed method tries to disentangle various speech factors (pitch, phoneme, loudness) based on speech processing models in the encoder, but this architecture has no experimental validation. I suggest the authors provide the contribution of these factors (possibly through the ablation study of each aspect) to the overall performance.”__
>
> We agree with the reviewer that additional ablation experiments could better illustrate the contributions of components of the proposed architecture. If accepted, we plan to perform ablations on the encoder module, as the bottleneck module is a simple recurrent network and the decoder module implements a variation of the filtering schemes proposed in [2] and [3]. This would include each sub-encoder (pitch/aperiodicity, loudness, phoneme predictor) and the fixed conditioning mechanism.
>
> That said, the choice of components is not unmotivated within the paper itself, as we note that our proposed architecture incorporates aspects of a number of experimentally-validated works. For example, the use of pitch, loudness, and residual spectrogram features is based on the work in [3], in which the authors also use a recurrent bottleneck to control a differentiable filter among other signal-processing components. Our choice of phonetic and pitch features also draws on [4], in which they are used to perform low-latency voice conversion. We have revised Section 3.1 (line 158) to clarify the former connection, while the latter is remarked on in line 147.
>
> __4.  “I have the same discussion for the loss function. There is no experimental justification for the proposed training scheme.”__
>
> While multi-resolution spectral losses are common in the speech-enhancement and generation literature [5], we agree with the reviewer that additional experiments could better illustrate the contribution of the choice of auxiliary loss function. While we could perform not all requested studies within the rebuttal period, if accepted, we would be willing to perform additional experiments evaluating auxiliary loss functions such as waveform-only regression losses, the MFCC cosine-distance loss of [3], and ASR feature-matching.

---

> > ### Author Response · Authors · 2022-08-01
> > **Response (continued)**
> >
> > __5. “Did you perform automatic speech recognition? I'm expecting that ASR would be robust against the ‘time-invariant’ FIR filter, and I'm curious how it would degrade the performance due to the proposed method based on the time-variant FIR filter.”__
> >
> > In the newly-added Appendix B3, we evaluate the intelligibility of audio produced by the proposed and baseline methods using ASR word error rate and character error rate metrics. Our experiments show that when targeting a speaker-recognition system, the proposed method results in word- and character-error rates 20+ times lower than pure signal-processing attacks of similar effectiveness (spectral gating and white noise) and lower than than the universal additive attack. This indicates our method's ability to preserve the intelligibility of speech while adversarially modifying speaker characteristics.
> >
> > We did not consider a time-invariant FIR filter attack in the paper because such an approach cannot adapt to incoming speech; in our discussion of related work, we note that the fixed-filter attack of [6] (Section 2, line 107) achieves lower success rates than traditional attacks on speaker recognition systems. See also our response to point 8 below.
> >
> > __6. “Is it possible to evaluate this method with the Voiceprivacy challenge benchmark? Then, the paper can compare various techniques proposed in the challenge or literature.”__
> >
> > Unfortunately, we were not able to evaluate our method under the VoicePrivacy benchmark [7] within the rebuttal period. While we would be open to adding such an evaluation were our manuscript to be accepted, we note that, as far as we are aware, no baseline systems or past entries to the VoicePrivacy challenge are designed to be human-imperceptible. The subjective and objective metrics used to evaluate entries consider naturalness and intelligibility, but not perceptual similarity to the original voice. Thus, despite the significant overlap between the datasets used for training and evaluation in our work and those used in the VoicePrivacy benchmark, a fair comparison against existing entries may be difficult due to this fundamental difference in objectives. Performing additional human subjects evaluations on existing entries may help in this regard, but the expense and effort required would be beyond the scope of this paper.
> >
> > __7. “‘VoiceBox’ is the same name as the famous (Matlab) speech processing toolkit. Although this is an old toolkit (based on Matlab!), I recommend the authors respect this prior work and avoid using this name.”__
> >
> > To avoid confusion with this prior work, our manuscript has been revised to change the name of our proposed system to “VoiceBlock.” If accepted, we will make corresponding changes to supplementary materials (website and code) before they are made publicly available.
> >
> > __8. “I would like to have more discussions on why the authors use the time-variant FIR filter with more scientific perspectives. I agree that a time-invariant FIR filter would be less distorted than the additive noise as it would just physically change the speaker or room characteristics. But I'm not very sure about the time-variant FIR filter.”__
> >
> > We refer to our discussion of point 5 above; additionally, we provide audio examples of our time-variant filter method via the online supplement (https://master.d3hvhbnf7qxjtf.amplifyapp.com/). The goal of our proposed method is to apply subtle, near-imperceptible filtering rather than to noticeably alter characteristics of the speaker’s channel. It is therefore crucial that the filter parameters be allowed to adapt to incoming speech audio, as at any given instant, the most effective and inconspicuous set of frequency-domain modifications may change. We note that we constrain filter parameters in an attempt to minimize perceptible fluctuations (see Section 3.3 and Appendix A.1 for details). Note, also, that time-varying filtering is common and typically goes unnoticed in the real world. Any indoor public venue with people moving about in it is an example. As people move, their bodies continually change the filtering experienced by the listener. Such changes typically go unremarked and unnoticed.
> >
> > That said, future work could very well explore more perceptible “fixed” perturbations that mimic realistic (if degraded) channel conditions (e.g. FIR reverb).

---

> > > ### Author Response · Authors · 2022-08-01
> > > **Response (continued, continued)**
> > >
> > > __9. “Can you discuss 56ms? I think this range would be accepted in many speech interface applications. For example, DNS challenges and other real-time speech processing tasks set a similar range for the latency constraint. However, some hearing devices set the latency constraint to less than 10 ms (you can check the Clarity challenge). This kind of discussion makes the paper more attractive since we can imagine the main application of this method.”__
> > >
> > > We refer to our response to point 2 above. Regarding the low-latency operating constraints of certain task environments, we note that our proposed system is intended to operate in digital communications contexts in which there is no “direct” unprocessed signal competing with the processed audio — for example, voice-over-IP telephony, videoconferencing, or live-streaming. In these contexts, a small fixed delay is often acceptable. By contrast, a speech enhancement algorithm operating in a hearing device would require much lower latency due to the simultaneous presence of processed and unprocessed signals in the listener’s perception.
> > >
> > > __10. “How much is the phoneme classifier part robust against the other languages, speaking styles, and noise environments? Librispeech ‘train-clean-100’ is English/clean/read speech, which would not cover many speech variations.”__
> > >
> > > We agree that the effectiveness of the phoneme classifier may vary depending on the user’s language, speaking style, and other factors; this variability in part motivated our design of the encoder, which also leverages pitch, aperiodicity, loudness, and spectrogram features. Unfortunately, we did not have sufficient time to evaluate the proposed method on non-English corpora. Moreover, evaluating the accuracy of the phoneme classifier beyond the LibriSpeech dataset would require the creation of additional aligned phoneme labels.
> > >
> > > However, we note that the VoxCeleb1 dataset contains utterances spanning a variety of languages, accents, and noise conditions. Our model’s strong performance on this dataset, as shown in Table 1, demonstrates a clear capability for generalization beyond clean English read speech. We consider it an encouraging sign that our model is trained on data drawn from a limited set of conditions, without data augmentation, and yet still demonstrates an ability to generalize to a much larger set of speech and channel conditions.
> > >
> > > Finally, we note that the focus on English-language speech is  widespread in both the speech de-identification and adversarial attack literature; the training and evaluation corpora used in the VoicePrivacy challenge, for example, consists of VoxCeleb 1-2 and the English-language LibriSpeech, LibriTTS, and VCTK datasets. While a number of recent works have investigated language dependence in speaker recognition and adversarial attacks, we feel this lies outside the scope of the paper.
> > >
> > > __11. “Can you discuss the RTF? Is 0.255 or 0.200 sufficiently light?”__
> > >
> > > As demonstrated by our streaming implementation, both RTF values are sufficiently light to allow smooth real-time operation on commodity laptops. For reference, the authors of the Demucs speech-enhancement model [5] deem a single-core RTF of 0.8 suitable for realistic use alongside video-conferencing software.
> > >
> > > Finally, we note that while we have leaned away from introducing extended discussions of latency and RTF (points 8, 9, 11) into our revised manuscript given the current length limit and focus of the paper, we would be open to incorporating these discussions in the extra page afforded in the camera-ready version were our paper to be accepted.
> > >
> > > We would like to reiterate our appreciation of the reviewer for their detailed comments, and we believe that the paper has been made stronger through our attempts to address them. Thank you!

---

> > > > ### Author Response · Authors · 2022-08-01
> > > > **References**
> > > >
> > > > References
> > > >
> > > > [1] Mia Chiquier, Chengzhi Mao, and Carl Vondrick. Real-time neural voice camouflage. In
> > > > International Conference on Learning Representations (ICLR), 2022.
> > > >
> > > > [2] Jesse Engel, Lamtharn Hantrakul, Chenjie Gu, and Adam Roberts. Ddsp: Differentiable digital signal processing. In International Conference on Learning Representations (ICLR), 2020.
> > > >
> > > > [3] Patrick O’Reilly, Pranjal Awasthi, Aravindan Vijayaraghavan, and Bryan Pardo. Effective and inconspicuous over-the-air adversarial examples with adaptive filtering. In International Conference on Acoustics, Speech and Signal Processing (ICASSP), 2022.
> > > >
> > > > [4] Damien Ronssin and Milos Cernak. Ac-vc: Non-parallel low latency phonetic posteriorgrams
> > > > based voice conversion. In IEEE Automatic Speech Recognition and Understanding Workshop (ASRU), 2021.
> > > >
> > > > [5] Alexandre Défossez, Gabriel Synnaeve, and Yossi Adi. Real time speech enhancement in the waveform domain. In Interspeech, 2020
> > > >
> > > > [6] Shimaa Ahmed, Yash Wani, Ali Shahin Shamsabadi, Mohammad Yaghini, Ilia Shumailov, Nicolas Papernot, and Kassem Fawaz. Pipe overflow: Smashing voice authentication for fun and profit. arXiv preprint arXiv:2107.14642, 2022.
> > > >
> > > > [7] Natalia Tomashenko, Xin Wang, Xiaoxiao Miao, Hubert Nourtel, Pierre Champion, Massimiliano Todisco, Emmanuel Vincent, Nicholas Evans, Junichi Yamagishi, and Jean-François Bonastre. The VoicePrivacy 2022 Challenge Evaluation Plan. arXiv preprint arXiv:2203.12468, 2022.
> > > >
> > > > [8] Szigeti, Tim and Hattingh, Christina. End-to-End QoS Network Design: Quality of Service in LANs, WANs, and VPNs. Cisco Press, 2004.
> > > >
> > > > [9] M. Kassim, R. A. Rahman, M. A. A. Aziz, A. Idris and M. I. Yusof, "Performance analysis of VoIP over 3G and 4G LTE network," 2017 International Conference on Electrical, Electronics and System Engineering (ICEESE), 2017, pp. 37-41, doi: 10.1109/ICEESE.2017.8298391.
> > > >
> > > > [10] https://www.verizon.com/business/terms/latency/

---

> > > > > ### Comment · Reviewer_NLhW · 2022-08-03
> > > > > **Thanks for the fruitful comments!**
> > > > >
> > > > > I enjoyed reading their responses, and all are very reasonable to me.
> > > > > I raised my score based on the responses and several additions made by the authors.
> > > > >
> > > > > I agree with you about the scope of the topic. Still, if we want to appeal to this technique, we may also think of other applications, i.e., can we apply the high-level concept of this approach to the additional time-series signal? The model architecture may need some modifications, but biosignal for health monitoring or driving signals for safe driving can be one application where we have a similar problem of de-identifying the user while preserving the important content information. I heard they also use similar signal processing techniques developed in speech/audio processing to some extent, and the time-varying filter idea might work. Sorry that I do not have clear ideas; this is a random thought.
> > > > >
> > > > > (Also, I apologize that I may paste a similar comment already. I'm a bit confused about the interface).

---

### Meta-Review · Area_Chair_ZJp9 · 2022-08-31

**Recommendation:** Accept
**Confidence:** Certain

**Metareview:**

The paper proposes an adversarial attack strategy, for audio-to-audio modeling, that preserves user’s privacy against speaker recognition models.

All reviewers agree this is a relevant topic for NeurIPS, with a strong contribution on voice privacy. In addition, everyone agrees the experimental section is solid (concerns have been addressed during the rebuttal phase).

**Award:**

No

---

### Decision · Program_Chairs · 2022-09-14

Accept